# Prediction of heterosis in the recent rapeseed (*Brassica napus*) polyploid by pairing parental nucleotide sequences

**Qian Wang**[1], **Tao Yan**[1], **Zhengbiao Long**[1], **Luna Yue Huang**[2], **Yang Zhu**[1], **Ying Xu**[1], **Xiaoyang Chen**[3], **Haksong Pak**[1], **Jiqiang Li**[4], **Dezhi Wu**[1], **Yang Xu**[5]*, **Shuijin Hua**[6]*, **Lixi Jiang**[1]*

**1** Institute of Crop Science, Zhejiang University, Hangzhou, China, **2** Department of Agricultural and Resource Economics, University of California, Berkeley, California, United States of America, **3** Institute of Crop Science, Jinhua Academy of Agricultural Sciences, Jinhua, China, **4** Institute of Crop Science, Zhangye Academy of Agricultural Sciences, Zhangye, China, **5** Agricultural College, Yangzhou University, Yangzhou, China, **6** Institute of Crop and Nuclear Agricultural Sciences, Zhejiang Academy of Agricultural Sciences, Hangzhou, China

* xuyang_89@126.com (YX); sjhua1@163.com (SH); jianglx@zju.edu.cn (LJ)

**Data Availability Statement:** All of the raw reads of the rapeseed accessions generated in this study have been deposited in the public database of the National Center of Biotechnology Information with

## Abstract

The utilization of heterosis is a successful strategy in increasing yield for many crops. However, it consumes tremendous manpower to test the combining ability of the parents in fields. Here, we applied the genomic-selection (GS) strategy and developed models that significantly increase the predictability of heterosis by introducing the concept of a regional parental genetic-similarity index (PGSI) and reducing dimension in the calculation matrix in a machine-learning approach. Overall, PGSI negatively affected grain yield and several other traits but positively influenced the thousand-seed weight of the hybrids. It was found that the C subgenome of rapeseed had a greater impact on heterosis than the A subgenome. We drew maps with overviews of quantitative-trait loci that were responsible for the heterosis (*h*-QTLs) of various agronomic traits. Identifications and annotations of genes underlying high impacting *h*-QTLs were provided. Using models that we elaborated, combining abilities between an *Ogu*-CMS-pool member and a potential restorer can be simulated *in silico*, sidestepping laborious work, such as testing crosses in fields. The achievements here provide a case of heterosis prediction in polyploid genomes with relatively large genome sizes.

## Author summary

Oilseed rape (*Brassica napus*) is of significant economic interest worldwide, providing high-quality oil with excellent health-promoting properties. It represents an excellent model of a successful recent polyploid that rapidly became an important crop worldwide. The utilization of hybridization, leading to hybrid vigor, or heterosis, is a successful strategy in increasing yield and vigor for many field crops including rapeseed (*Brassica napus*). However, the procedure of using classical breeding methods remains slow and laborious,

BioProject ID PRJNA664250. Other relevant data are within the manuscript and its Supporting Information files.

**Funding:** LJ received grants (No, 32130076, 31961143008) from the National Natural Science Foundation of China (http://www.nsfc.gov.cn/), and a grant (No.2021C02057) from Zhejiang Provincial Key Research Projects (http://kjt.zj.gov.cn/). XC received a grant (2020ZY1019) from Local Scientific and Technological Development Projects Guided by Chinese Academy of Agricultural Sciences (http://kjj.jinhua.gov.cn/). The cost of the study was covered by the above grants. The funders had no roles in the study design, data analysis, decision to publish, or preparation of the manuscript.

**Competing interests:** The authors have declared that no competing interests exist.

illustrating the need for predictive and innovative methods. Here, we have achieved a significant breakthrough by using genome selection and significantly advanced models to predict the heterosis by pairing genome-wide nucleotides of parents. We provided maps with overviews of quantitative trait loci that were responsible for the heterosis of various agronomic traits. The research used deep resequencing (>30x) data of the entire polyploidy rapeseed genome, providing a successful case for the prediction of heterosis in polyploid genomes with relatively large genome sizes. Moreover, we provided the genetic information (SNPs) of 1007 core accessions of this species in the public domain for testing combinations with high heterosis using our predicting model for rapeseed breeders all over the world.

## Introduction

Heterosis, which is a product of crossing two parents with different genetic backgrounds, is a common phenomenon in the biological world. The hybrid generation often displays more vigor, greater resistance to disease, better adaptability under stressful environments, and higher yield, when compared with the parents. Heterosis was first discovered in a tobacco hybridization experiment approximately 150 years ago, and it has been applied extensively for yield improvement in various field crops such as rice [1], corn [2], cotton [3], rapeseed [4], and some vegetables [5].

High-parent heterosis (HPH) and mid-parent heterosis (MPH) describe the degrees of phenotypic differences between a hybrid and its better parent and between a hybrid and the average of its two parents, respectively [6]. Numerous theories have been used to explain heterosis, and the major ones are the dominance and over-dominance hypotheses. The dominance hypothesis attributes the enhanced performance of hybrids to the repression of undesired recessive alleles of a parent by dominant favorite alleles of the other parent, and the poor performance of inbred lines to the loss of a diverse genetic basis, which is manifested by numerous homozygous loci [7]. Conversely, the over-dominance hypothesis attributes the superiority of heterozygotes to the survival of alleles that are recessive and harmful in homozygotes, and the poor performance of inbred lines to high proportions of such deleterious recessive alleles [8].

Dominance and over-dominance effects give rise to different gene expression profiles in offspring. Considering over-dominance is the main source of superiority in adaptability under heterosis, certain genes in heterozygous individuals could be overexpressed in comparison to their homozygous parents. However, in the case of dominance, fewer genes would be downregulated in heterozygous individuals compared with their parents. Based on such assumptions, greater heterosis would be generated with an increase in heterozygous loci.

To obtain ideal hybrids, breeders have to generate high numbers of hybridization combinations and test their performance under multiple environments over time. Genomic selection (GS), a novel approach in which selection is not performed based on a few markers but on a genome-wide marker dataset, combines marker data with phenotypic and pedigree data (if available), and attempts to accurately predict the performance of the next generation rather than to identify individual loci that are significantly associated with a trait, with more rapid results and reduced costs in breeding activities. In addition, GS, which considers the entire genome sequences of parents as valuable breeding assessments and captures single-site effects even if they are minimal, can shorten the breeding cycle considerably, and save a lot of time and labor. At present, the major methods of developing GS include the genomic best linear unbiased prediction (GBLUP) method [9], and the least absolute shrinkage and selection

operator (LASSO) method [10]. The Pearson correlation coefficient between the observed and predicted phenotypic values is often an indicator of the prediction ability [11,12].

Rapeseed (*Brassica napus*), a typical amphidiploid species, which originated from interspecific hybridization between *Brassica rapa* (AA, n = 10) and *Brassica oleracea* (CC, n = 9) only 7500 years ago [13], is one of the significant economic interests worldwide, providing high-quality oil with excellent health-promoting properties, and with a significant potential for non-food use such as biofuels and bioplastics. The yield and overall production of the crop have been increased significantly owing to the commercial use of hybrids in major rapeseed production areas, such as Canada, China, and Europe. As for many other crops, the strategy of choice for large-scale commercial production of hybrids relies on the development of cytoplasmic male-sterile parents, which fertility can be restored when crossed to another parent carrying a restorer-of-fertility gene. In rapeseed, the *Polima* cytoplasmic male sterile (*Pol*-CMS) system is widely used in semi-winter Chinese ecotype breeding [14]. Although it is relatively easy to find restorers for *Pol*-CMS lines, male sterility in the *Pol*-CMS type lines is unstable under certain environmental conditions. However, the *Ogura* Cytoplasmic Sterile (*Ogu*-CMS) is much more stable and complete; nevertheless, finding restorers for *Ogu*-CMS lines is challenging, and it takes several years to transfer restoring genes into potential restorers.

In this study, we successfully bred a series of *Ogu*-CMS restorers and constructed a pool of *Ogu*-CMS lines, which reflects the genetic diversity of the semi-winter ecotype [15]. We applied GS and developed models to predict the heterosis by pairing genome-wide nucleotides of parents. Maps with an overview of quantitative trait loci for heterosis (*h*-QTLs), at which parental genetic similarity index (PGSI) positively or negatively correlated with the heterosis of a specific trait, were drawn. With the GS-based predictive models that we elaborated, combining abilities between an *Ogu*-CMS-pool member and one of 1007 potential restorers could be tested *in silico* by pairing the nucleotide sequences of parents. This will fasten rapeseed breeding by saving years of effort, and provide a case of study of heterosis prediction in polyploid genomes with relatively large genome size.

## Results

### Heterosis of F1 hybrids of the *Ogu*-CMS system

We developed an *Ogu*-CMS pool consisting of 50 members in addition to eight *Ogu*-restorers for this experiment. The identifications (ID) and relevant information of the CMS and restorer lines are provided in S1 Table. The 50 *Ogu*-CMS lines constituted an *Ogu*-CMS pool that had a wide genetic diversity reflected by a principal component analysis (PCA) based on 1,057 sequenced genomes, including those of the CMS lines used in the present study. The lines represent the semi-winter ecotype in the background of a worldwide germplasm collection (S1 Fig) [15].

Crosses between the 50 CMS lines and the eight restorers yielded 400 hybrids. The hybrid lines were grown under three environmental conditions from 2017 to 2019. They demonstrated significant heterosis in terms of HPH and MPH across various agronomic traits such as plant height (PH), the number of seeds per silique (NSS), and grain yield (GY). The phenotypic data and the genetic relationship between parents and offspring are provided in S2 and S3 Tables. In addition, they exhibited significant MPH across traits such as number of branches per plant (NBP), number of siliques per plant (NSP), and thousand-seed weight (TSW) (Fig 1). GY-HPH and GY-MPH values of the top 10% hybrid lines were 90.16% and 146.33%, respectively, whereas, the GY-HPH and GY-MPH values of the top 1% hybrid lines were as high as 168.81% and 233.01%, respectively (S4 Table).

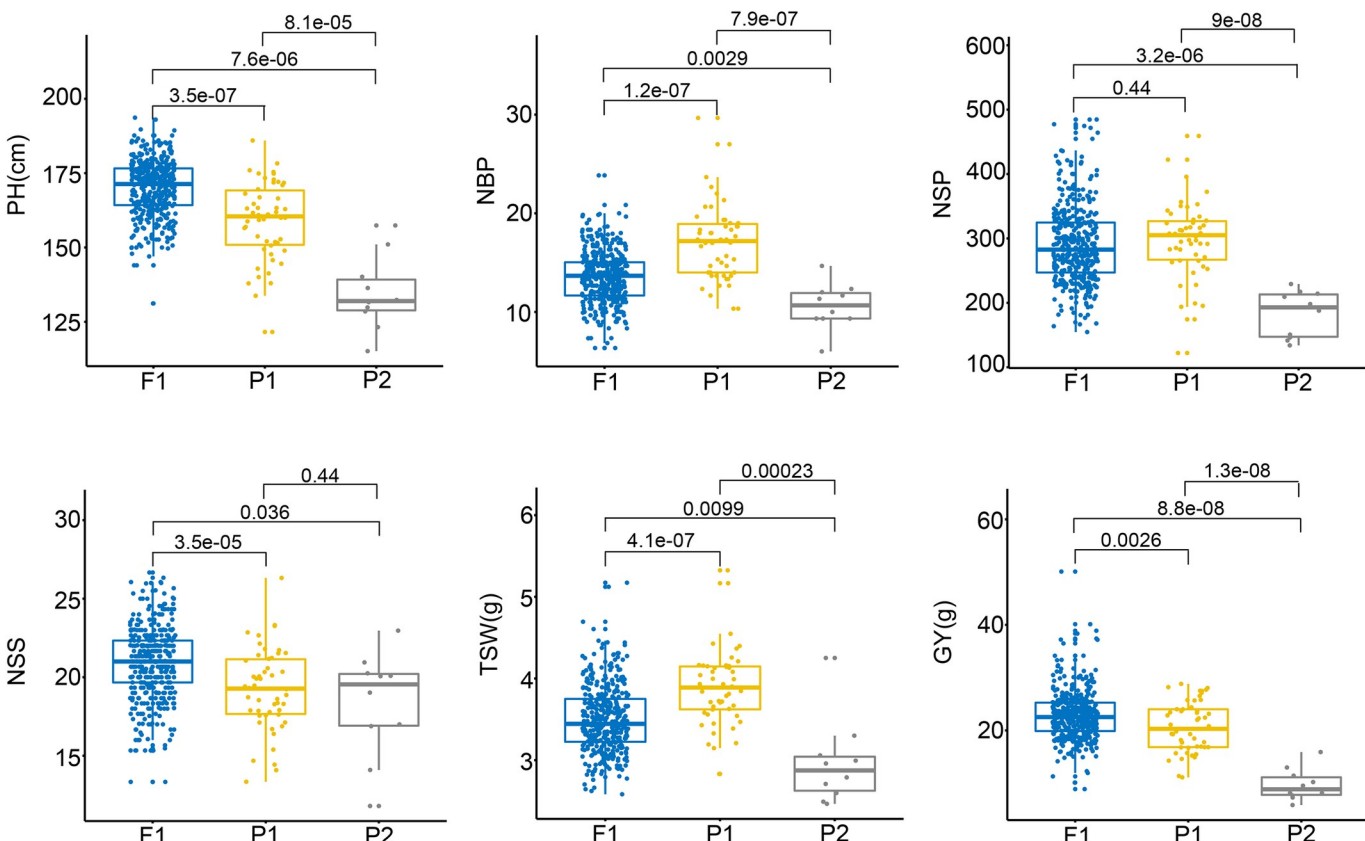

**Fig 1. Comparison of agronomic-trait performance between the F1 hybrids and their parents.** PH: plant height, NBP: number of branches per plant, NSP: number of siliques per plant, NSS: number of seeds per silique, TSW: thousand-seed weight, and GY: grain yield. P1 represents female parents, and P2 represents male parents. The values indicate the significance of pairwise comparisons.

To estimate the influence of parents on heterosis, we calculated the correlation coefficients between the phenotypic values of the parents and those of the hybrids. For most traits, the correlations between the hybrids and their male parents were higher than those between the hybrids and their female parents (S5 Table). Notably, the correlation coefficient of PH between the hybrids and male parents was the highest (r = 0.64), indicating a higher impact of the male parents on the PH of the hybrids. Furthermore, we compared the correlations among the six agronomic traits of the hybrids. There were relatively high positive correlations between GY and NSP (r = 0.63), NSP and NBP (r = 0.46), NSS and PH (r = 0.31), and relatively high negative correlations between TSW and NSS (r = -0.31), TSW and NSP (r = -0.15), and NSP and NSS (r = -0.21) (Fig 2B and S6 Table). Among the six traits, NSP had the greatest correlation with GY, suggesting a considerable influence of silique number on yield heterosis in rapeseed (Fig 2A). Overall, the heterosis of the hybrids in the semi-winter ecotype with the *Ogura* system was significant and attractive.

## Correlation between parental genetic similarity index (PGSI) and F1 heterosis

To determine the mechanism by which parental-sequence similarity potentially influences hybrid vigor, we calculated the correlations between the genome-wide PGSI and heterosis. A total of 4.44 million SNPs were obtained across the paring genomes by mapping reads to the

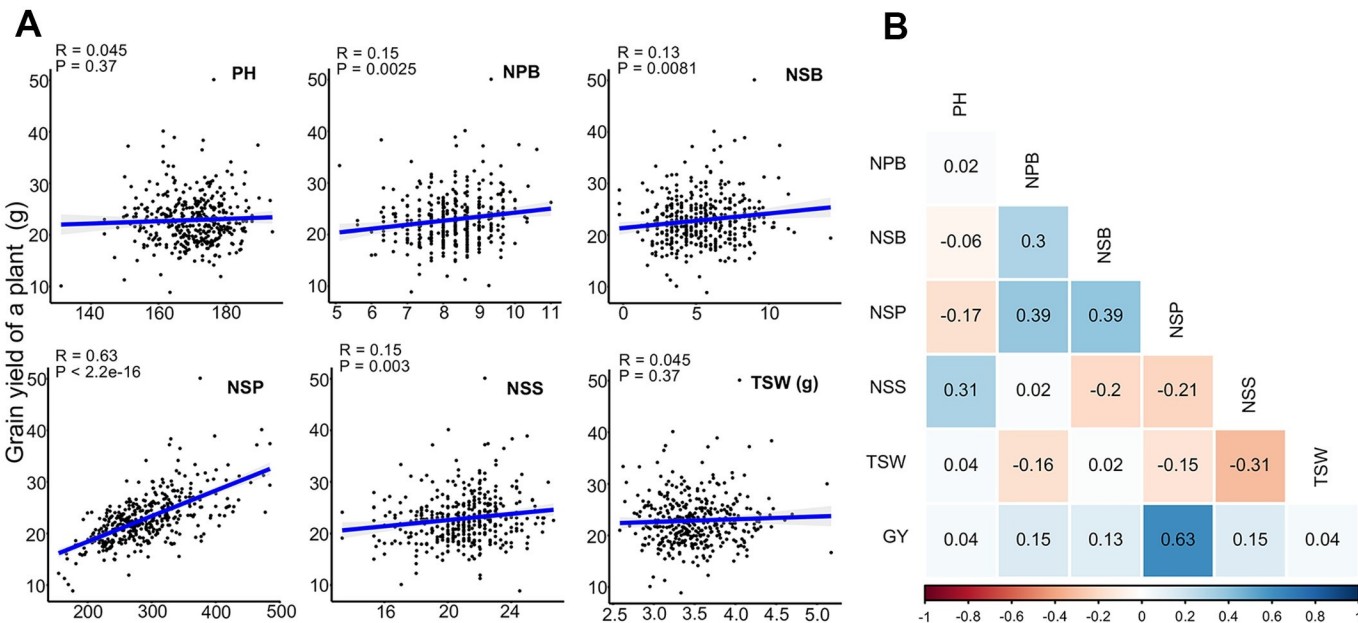

**Fig 2.** Correlations between the phenotypic value of agronomic traits (left) and the contribution of a trait to grain yield (right). PH: plant height, NBP: number of branches per plant, NPB: number of primary branches per plant; NSB: number of secondary branches; NSP: number of siliques per plant, NSS: number of seeds per silique, TSW: thousand-seed weight, GY: grain yield. (**A**) Contribution of a specific trait to the grain yield. R represents the correlation coefficient and P represents significant values. (**B**) Pairwise correlations among the phenotypic values of agronomic traits. The sectors indicate the positive or negative values of the correlations. The darker the sectors, the greater the absolute values. The number inside the box represents the correlation coefficient. Changes in color from dark red to dark blue correspond to changes in correlation coefficient from -1 to +1.

reference genome [13]. Overall, PGSI negatively influenced GY, NSS, NSP, and PH, but positively influenced TSW, regardless of the type of heterosis (HPH and MPH) (Fig 3). The absolute values of the correlation confidence between PGSI and NSS-HPH, TSW-HPH, PH-MPH, and NSS-MPH were relatively high (S7 Table).

As rapeseed (*Brassica napus*) is a typical polyploid species, we compared the influence on heterosis between the A and C subgenomes. In general, the influence from the C subgenome was greater than the influence from the A genome on heterosis across the six traits. Furthermore, we compared the influences between the 19 chromosomes making up the whole genome. The PGSIs of C01, A03, C05, C06, C02, A06, C09, A01, C04, A07, C08, A02, A09, A04, C03, A10, C07, A05, and A08 had influences (in order from high to low) on HPH, respectively. The PGSIs of C05, A03, C04, C01, C03, A07, C09, C02, C06, A09, A01, A06, A10, C08, A02, A05, A04, A08, and C07 had influences (in order from high to low) on MPH, respectively (S8 Table). Here, the influence was calculated by stacking up the absolute values of the correlation coefficients, where positive correlations meant that the higher the PGSI, the smaller the heterosis, negative correlations indicated that the lower the PGSI, the greater the heterosis. The absolute value indicates the degree of impact.

## Genomic regions where parental genetic similarity impacts on heterosis of the *Ogura* hybrids

We calculated PGSI and performed LASSO analysis to identify the genomic regions responsible for the heterosis of traits, which were defined as *h*-QTLs. 172 *h*-QTLs were associated with GY-HPH (Fig 4 and S9 Table). Some *h*-QTLs had a relatively greater effect on heterosis, as shown with the darker colors in Fig 4. The darker the colors of circles or triangles, the greater

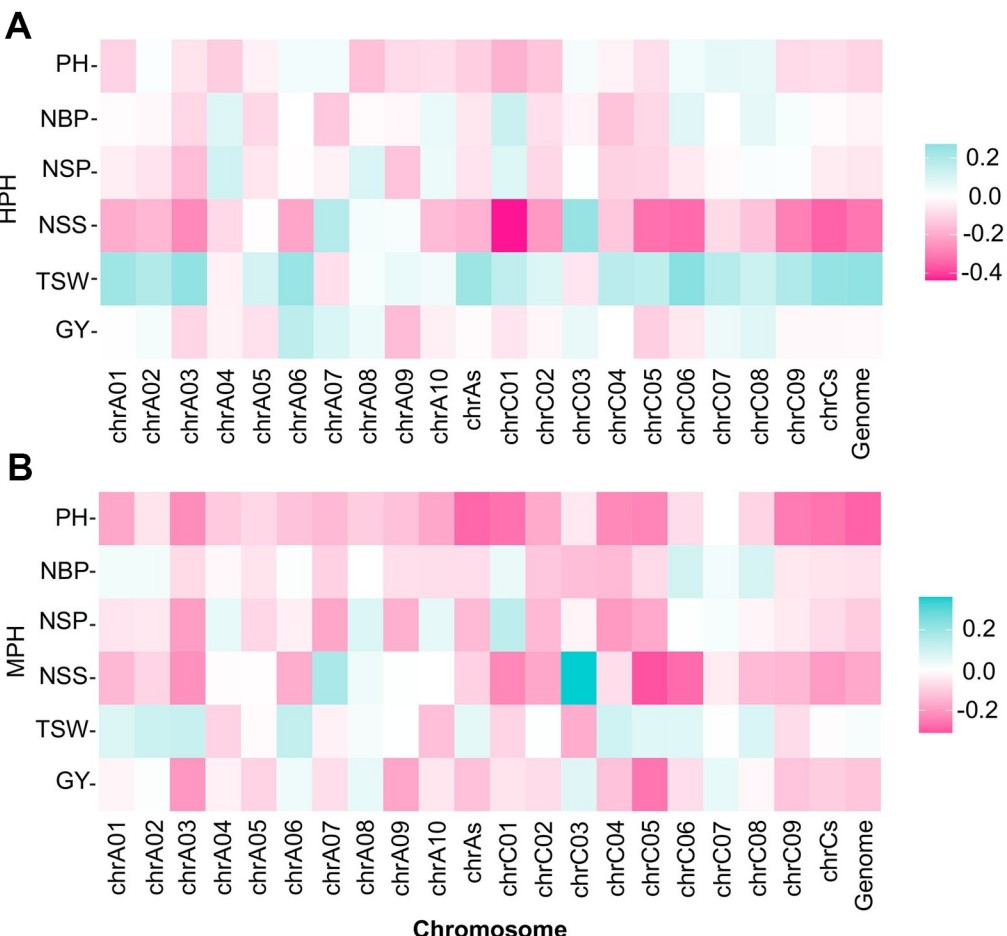

**Fig 3. Effects of parental genetic similarity index (PGSI) on heterosis of traits.** The pink and green squares represent negative and positive effects, respectively. The darker the squares, the larger the absolute value of the correlation coefficients. (**A**) Effect of PGSI on high-parent heterosis (HPH). (**B**) Effect of PGSI on mid-parent heterosis (MPH).

the impacts on heterosis, either positive or negative. We also observed 130, 60, 102, 111, and 104 *h*-QTLs associated with TSW-HPH, NSS-HPH, NSP-HPH, NBP-HPH, and PH-HPH, respectively (S2–S6 Figs and S10–S14 Tables). Consistent with the results illustrated in Fig 3, the C subgenome had a higher impact on the heterosis, accounting for 57.0%, 56.6%, 62.7%, 62.7%, 63.1%, and 62.5% of the *h*-QTLs responsible for GY-HPH, TSW-HPH, NSS-HPH, NSP-HPH, NBP-HPH, and PH-HPH, respectively.

In the present study, all the variables in the LASSO prediction model were defined as *h*-QTLs, and the top 10% and the bottom 10% regression coefficients of the variables were defined as high-impact *h*-QTLs. All *h*-QTLs for a specific trait were displayed on the maps and the underlying genes responsible for high-impact *h*-QTLs for investigated. There were 34 high-impact *h*-QTLs for GY-HPH according to the definition. The more heterozygous the *h*-QTLs such as Chr.C06-04, Chr.C08-01, the higher the GY-HPH. Conversely, the more homozygous the *h*-QTLs, such as Chr.C08-02 and Chr.C03-08, the higher the GY-HPH. The candidate genes covered by the high-impact *h*-QTLs for GY-HPH, PH-HPH, TSW-HPH, NSS-HPH, NSP-HPH, and NBP-HPH are listed in S15–S20 Tables.

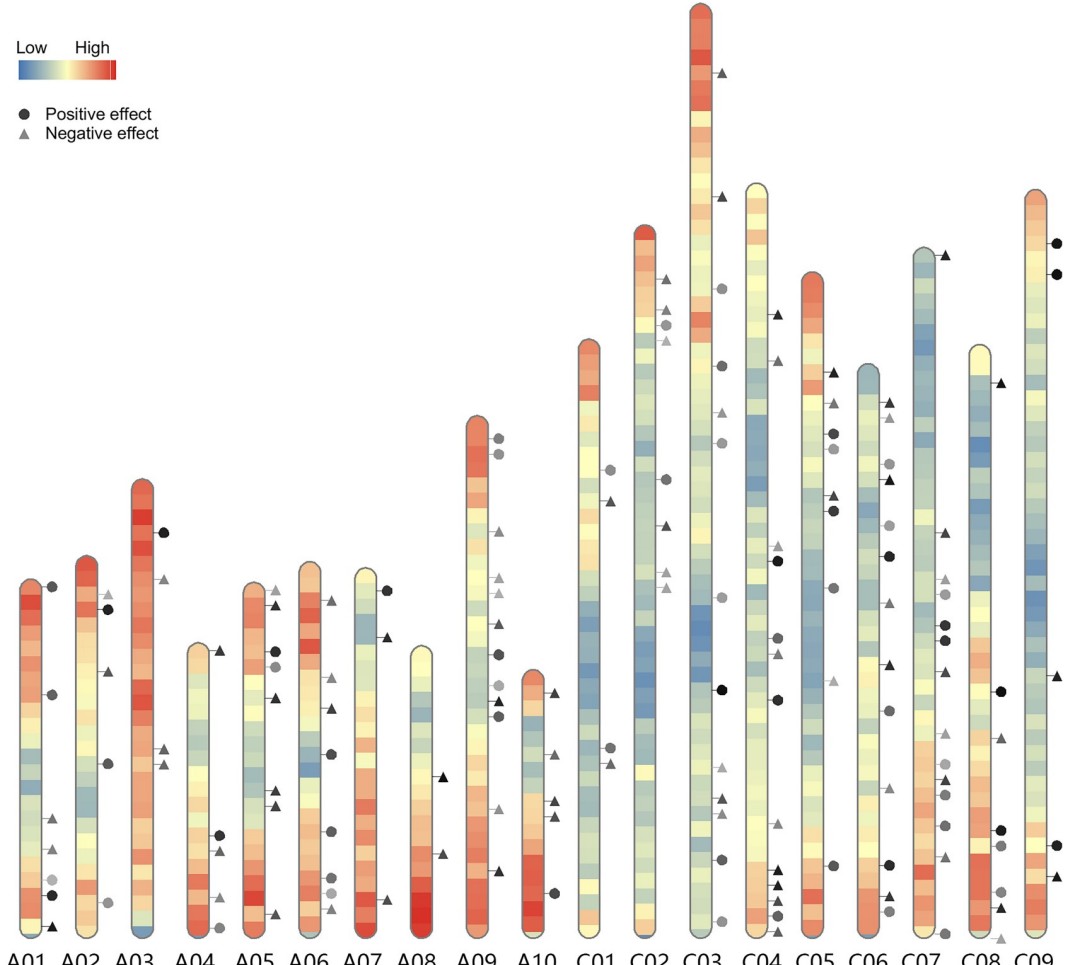

**Fig 4. *h*-QTLs responsible for GY-HPH.** The circles represent the *h*-QTLs that positively contributed to the GY-HPH, and the triangles represent *h*-QTLs that were negatively correlated to GY-HPH. The darker the colors of circles and triangles, the greater the effects of the *h*-QTLs, either positive or negative. The colors on the chromosomes indicate the density of genes. The darker the blue, the lower the gene density, the darker the red, the higher the gene density. A and C stand for the two sub-genomes of *Brassica napus*. A limited number of *h*-QTLs on randomly piled contigs, whose positions on certain chromosomes were unknown, are not shown on the map. A positive effect indicated with a circle on maps means the smaller the PGSI, the great the heterosis, whereas, a negative effect tagged with a triangle means the bigger the PGSI, the greater the heterosis.

**Table 1. Comparison of the predictability of high-parent heterosis (HPH) of six traits among six models.**

| Methods | GY | TSW | NSS | NSP | NBP | PH |
|---|---|---|---|---|---|---|
| GBLUP_A | 0.7165 | 0.7110 | 0.6686 | 0.7218 | 0.7543 | 0.8815 |
| GBLUP_AD | 0.7165 | 0.7251 | 0.6855 | 0.7218 | 0.7576 | 0.8834 |
| LASSO_SNP | 0.8000 | 0.7987 | 0.7616 | 0.8062 | 0.8246 | 0.9165 |
| LASSO_1Mb | 0.8828 | 0.8671 | 0.7990 | 0.8260 | 0.8697 | 0.9370 |
| LASSO_500Kb | 0.8649 | 0.8842 | 0.8516 | 0.8544 | 0.8745 | 0.9646 |
| LASSO_100Kb | 0.8379 | 0.8729 | 0.8837 | 0.8816 | 0.9246 | 0.9754 |

Notes: PH, plant height; NBP, number of branches per plant; NSS, number of seeds per silique; NSP, number of siliques per plant; TSW, thousand-seed weight; GY, grain yield.

**Table 2. Comparison of the predictability of mid-parent heterosis (MPH) of six traits among six models.**

| Methods | GY | TSW | NSS | NSP | NBP | PH |
|---|---|---|---|---|---|---|
| GBLUP_A | 0.6310 | 0.6463 | 0.7193 | 0.7136 | 0.6236 | 0.7808 |
| GBLUP_AD | 0.6310 | 0.6506 | 0.7230 | 0.7136 | 0.6276 | 0.7811 |
| LASSO_SNP | 0.7416 | 0.7550 | 0.8000 | 0.8000 | 0.7416 | 0.8485 |
| LASSO_1Mb | 0.7733 | 0.8071 | 0.8698 | 0.8199 | 0.8374 | 0.8928 |
| LASSO_500Kb | 0.7898 | 0.8134 | 0.9203 | 0.8342 | 0.8117 | 0.9240 |
| LASSO_100Kb | 0.7902 | 0.8352 | 0.8806 | 0.8670 | 0.8229 | 0.9490 |

Notes: PH, plant height; NBP, number of branches per plant; NSS, number of seeds per silique; NSP, number of siliques per plant; TSW, thousand-seed weight; GY, grain yield.

## Prediction of heterosis in a training population containing 400 hybrids via cross-validation

To predict heterosis of the hybrids with the *Ogu*-CMS system, we performed ten-fold cross-validation with 100 replicates in a training population containing 400 hybrids produced by 50 *Ogu*-CMS lines and eight restorers. To identify the optimal model for prediction, we compared the predictabilities of various models, namely GBLUP_A, GBLUP_AD, LASSO_SNP, LASSO_100Kb, LASSO_500Kb, and LASSO_1Mb. Parameters from ANOVA for all the predictions were listed in S21 Table. All predictabilities were greater than 0.6 (Table 1). Predictability varied across traits. For example, PH was the most predictable trait across all models. In addition, the predictability of HPH was higher than that of MPH across most traits, excluding NSS (Table 2).

Generally, the LASSO models demonstrated higher predictabilities than the GBLUP models. The GBLUP_AD model did not exhibit significantly higher predictability than the predictability of the GBLUP_A model. Among the four LASSO models, LASSO_SNP had the lowest predictability values across the six traits, regardless of the heterosis definition (HPH or MPH), indicating the necessity for reducing dimension in the calculations. In terms of HPH, the optimal model for NSS, NSP, NBP, and PH was LASSO_100Kb, and the optimal models for GY and TSW were LASSO_1Mb and LASSO_500Kb, respectively (Table 1). In terms of MPH, the optimal model for GY, TSW, NSP, and PH was LASSO_100Kb, and the optimal models for NSS and NBP were LASSO_500Kb and LASSO_1MB, respectively (Table 2). Overall, according to the results, an appropriate model should be selected to predict the heterosis of a specific trait. The LASSO_100Kb model was acceptable for the prediction of heterosis in all six traits (Fig 5).

## Further validation of heterosis prediction models

We adopted the LASSO_100Kb, LASSO_500Kb, and LASSO_1Mb models to predict the heterosis of a 100-hybrid population generated by the *Ogu*-CMS-pool members and two independent restorers. The predicted and actual values observed in fields were analyzed to determine the predictability (Fig 6 and S22 Table). For HPH, the correlation coefficients between the predicted and actual values were 0.84, 0.68, 0.66, 0.52, 0.64, and 0.65 for PH, NBP, NSP, NSS, TSW, and GY, respectively. For MPH, the correlation coefficients between the predicted and actual values were, conversely, 0.73, 0.36, 0.51, 0.62, 0.61, and 0.41 for PH, NBP, NSP, NSS, TSW, and GY, respectively. The predictability of PH was the highest, regardless of heterosis definition. As illustrated by the red and blue colors in Fig 6, the model could successfully indicate a superior restorer based on the performances of some certain traits. For example,

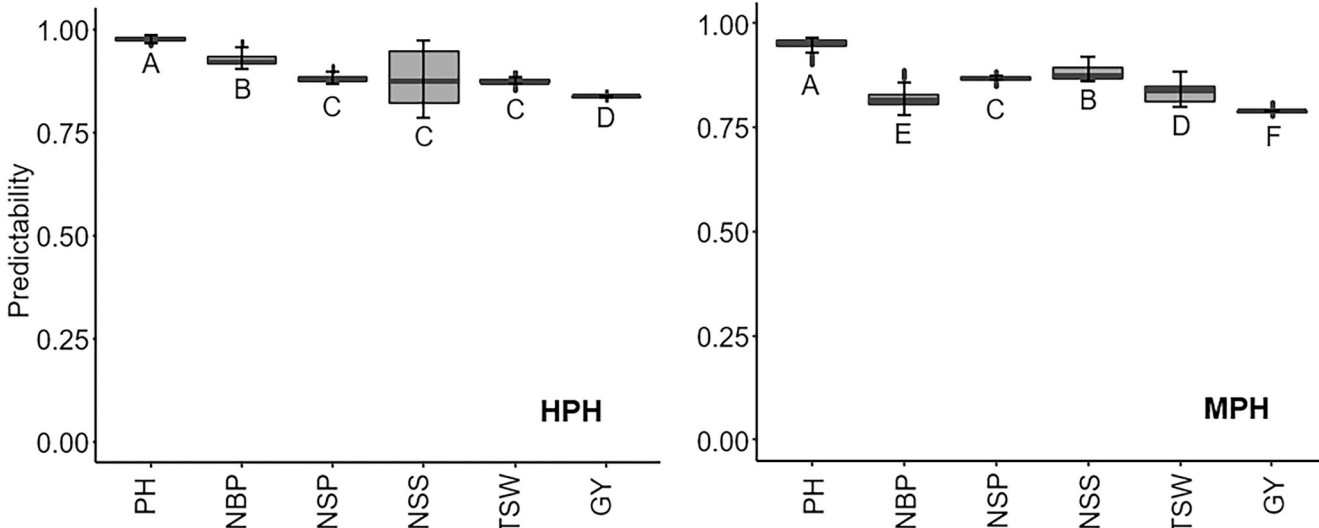

**Fig 5. Comparison of predictability of LASSO_100Kb model for high-parent heterosis (HPH) and mid-parent heterosis (MPH) among six agronomic traits.** Different letters indicate a significant difference (*p* = 0.01) between a comparison pair. PH: plant height, NBP: number of branches per plant, NSP: number of siliques per plant, NSS: number of seeds per silique, TSW: thousand-seed weight, and GY: grain yield.

Restorer No. 9 was superior to Restorer No. 10 in PH-HPH and TSW-HPH; conversely, Restorer No. 10 was superior to Restorer No. 9 in NSS-HPH. However, the GY-HPH, NBP-HPH, and NSP-HPH depended on specific combinations between the restorers and the *Ogu*-CMS-pool members (S1 Table), and it was hard to tell which restorer was better in yielding higher GY-HPH, NBP-HPH, and NSP-HPH. Nevertheless, the combinations for the highest GY-HPH, NBP-HPH, and NSP-HPH could be recommended based on the result.

## Discussion

As a promising new breeding method, GS has been applied to the prediction of heterosis of various crops such as rice [11,12,16], corn [17], barley [18], wheat [19], ryegrass [20], and pumpkin [21]. The traits which were predicted in the different studies were not only limited to yield and yield components [16–21], but also those traits such as biotic- and abiotic-stress tolerances [22,23], nutrient utilization efficiency [24,25].

Compared with the previous GS studies on other crops, our research had the following characteristics. First, the genetic information used in our study involves 4.44 million SNPs, which were 2–3 orders of magnitude more than the number of molecular markers used in the previous studies [16–25]. The previous studies either involved resequencing data of the crops with much smaller genomes such as rice [11,12,16], or only a small part of genome-wide SNPs of the crops with larger genomes such as barley [18] and corn [17]. We used deep resequencing (>30x) data of the entire polyploidy rapeseed genome, providing a successful case for the prediction of heterosis in polyploid genomes with relatively large genome sizes. More SNP markers tend to imply a more comprehensive level of genome coverage and indicate the involvement of more genetic information. However, a higher number of SNPs does not always mean higher predictability. Previous studies showed that the accuracy of prediction increases with more molecular markers within a specific region, but it reaches a peak after which increasing the density of markers is no longer beneficial for prediction accuracy [26–31]. In addition, there is a relationship between the number of markers required and the degree of linkage disequilibrium (LD) of the species. The more rapidly a species declines in LD, the

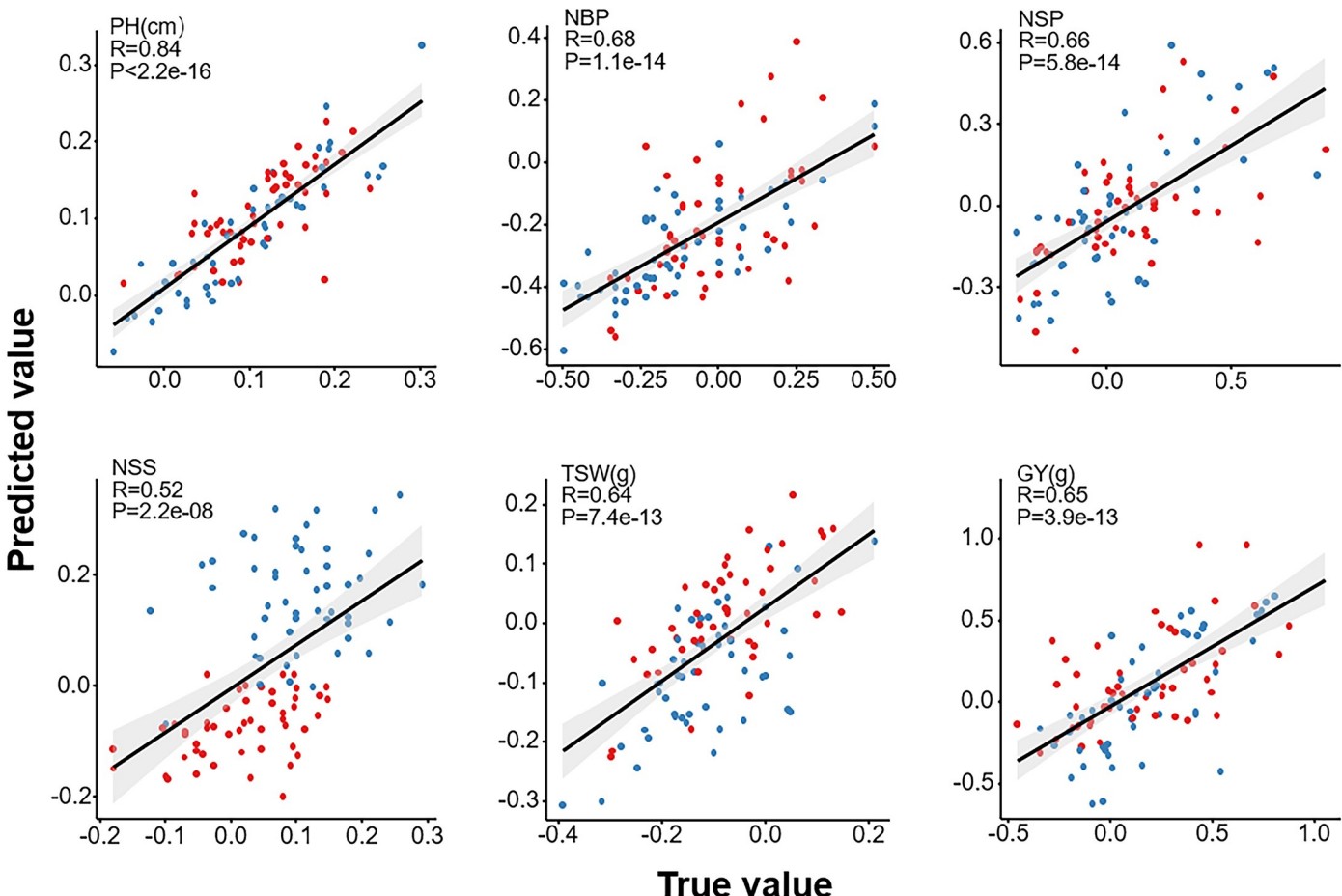

**Fig 6. Fitness between the predicted high-parent heterosis (HPH) and the actual observed HPH of the testing population containing 100 hybrid lines.** Each dot represents one of the 100 hybrid lines that arose from a cross between an *Ogu*-CMS-pool member and two restorers independent from the training population. The red and blue colors represent restorer 9 and 10, respectively. The grey areas indicate 95% confidence intervals.

smaller the LD distance, and more molecular markers would be required within the same size of the chromosomal fragment. The LD distances of the populations used in our study were less than 100Kb (S7 Fig), which is consistent with the previous study [15].

Second, instead of the direct use of SNPs, we introduced the concept PGSI to drastically reduce the dimension of the calculation matrix. The predictability was, therefore, increased to 0.8828 for GY-HPH and even 0.9754 for PH-HPH, which were much higher than those in previous reports were [16–24]. When we calculated the PGSI, we simply divided the chromosomes into 1Mb-, 500Kb-, and 100Kb-sizes, respectively, without considering local LD and gene numbers within the regions, which was technically complicated. LD is a concept of population genetics, meaning the non-random association of alleles at different loci. It is meaningful to calculate LD for a given population but is practically difficult to calculate PGSI based on LD because the LD distance of two parental genomes does not always match with each other, and an LD distance for an individual genotype was rather difficult to be determined.

Third, we created the concept of *h*-QTLs and drew maps with overviews of *h*-QTLs across the genome, at which PGSI positively or negatively associated with the heterosis as we showed for several agronomic traits. The *h*-QTL maps published in our study predict that the genomic regions (with a dark color) would exert a high impact on trait-specific heterosis, meaning that

the PGSI of those regions strongly correlates with the heterosis. We would suggest that those regions be very much considered in a GS breeding for heterosis. Since the LD distance of the rapeseed population is less than 100kb (S7 Fig), and an *h*-QTL of 100 Kb in size might span across two LD fragments on average. There must be a few genes that were responsible for a kind of heterosis despite the 'false'-gene majority (S15–S20 Tables). Most allelic changes might not associate with trait-specific heterosis. However, it is beyond the scope of this paper to identify the major functional genes of each *h*-QTL.

Hybrid vigor was well demonstrated here in semi-winter rapeseed ecotype based on the *Ogu*-CMS system, with the top 10% hybrids displaying 90.2% GY-HPH on average. Among the major yield components, NSP had the greatest correlation with GY, which is consistent with previous findings [4]. TSW was negatively correlated with NSS and NSP, and NSP was negatively correlated with NSS, since the traits, which are both limited by photosynthate allocation, counteract each other. Except for PH value, higher values of the traits such as TSW, NSS, NSP, and NBP were better for GY-HPH. Higher PH may give rise to higher biomass, which would positively affect yield. However, higher PH would not be always better for GY-HPH, e.g. lodging caused by plant height could negatively influence the final yield. In the present study, there was no correlation between PH-HPH and GY-HPH, which could be attributed to relatively low lodging during the seasons when the field experiments were conducted. The result shows that the male parents would affect the hybrids more than the female parents on the height of F1 plants. One of the possible reasons could be the much smaller size of the restorer-line (male) population than that of the CMS-line (female) population. The results of our study showed that a high overall PGSI would lead to high TSW; conversely, a low PGSI would favor a high NSS. To our knowledge, these interesting findings were not reported in other field crops.

The commercial use of the *Ogu*-CMS system for the semi-winter rapeseed ecotype would be a breakthrough in rapeseed production in the Yangtze River Basin because of the advantages of this CMS system in the form of stability and complete male sterility [32]. Most rapeseed genotypes can serve as *Ogu*-CMS line maintainers; however, the process of breeding an *Ogu*-restorer is slow and laborious [33]. In the present study, we established a pool of 50 *Ogu*-CMS lines, which represents the genetic diversity of the semi-winter rapeseed ecotype (S1 Fig). Models that could predict the heterosis of F1 plants from the crosses between the *Ogu*-CMS-pool members and potential restorers were developed. Using the models, we could test the combining ability between an *Ogu*-CMS-pool member and any potential restorer by pairing the nucleotide sequences *in silico*. Such a tool could bypass otherwise arduous manual work such as testing crosses in the field. Breeding efforts could be, therefore, focused on the transfer of restorer genes to a limited number of candidates, which are often achieved by backcrossing processes that usually take several years. To facilitate such applications, we established BnaSNPDB, an interactive web portal for efficient retrieval and analysis of Single Nucleotide Polymorphisms (SNPs) of 1,057 rapeseed germplasm accessions (https://bnapus-zju.com/bnasnpdb) [34]. SNPs of a genotype can be easily retrieved for *in silico* pairing with SNPs of an *Ogu*-CMS-pool member to simulate hybrid vigor. To validate the accuracy of the model, we created a test population using the same 50 *Ogu*-CMS lines and two independent restorer lines. The results of the present study show that different models are suitable for predicting different traits or heterosis based on respective definitions. For example, the LASSO_100Kb model is suitable for predicting NSS-HPH, NSP-HPH, NBP-HPH, and PH-HPH, whereas the LASSO_500Kb and LASSO_1Mb models are suitable for predicting TSW-HPH and GY-HPH. PH heterosis predictability was the highest among the six agronomic traits explored since PH could be more accurately measured than other traits, in which the errors were more challenging to control. Numerous studies have compared the predictability of heterosis across various

models. However, debate persists regarding the optimal method for predicting heterosis [11,16,35].

In the present study, the four LASSO methods were superior to the two GBLUP methods in terms of heterosis predictability. LASSO regression was characterized by variable selection and regularization of complexity while fitting the generalized linear model. Variable filtering is essential for LASSO, which means not inputting all variables into the model for fitting, but selectively inputting variables into the model to obtain better performance parameters. Complexity adjustment controls the complexity of the model through a series of parameters to avoid overfitting. For a linear model, complexity is directly related to the number of variables in the model. The more the variables, the higher the model complexity. Although including more variables in the fitting could often lead to a superior model, there is a risk of overfitting. In general, overfitting is possible when the number of variables is much greater than the number of data points, or when a discrete variable has too many unique values. In our study, there was no significance between LASSO_A and LASSO_AD in predicting the heterosis. This may be because the kinship matrix of additive effect has already captured much information about the kinship matrix of dominant effect. The dominance effect did not, therefore, play a significant role in accounting for the rest of the variances. The least predictability of the LASSO_SNP model could have arisen from the fact that we used SNP markers as variables. Too numerous variables could have led to overfitting, which, in turn, minimized heterosis predictability.

In general, there is a higher degree of genetic diversity in the A subgenome than in the C subgenome in large genetic populations, which might be caused by the fact that the A subgenome integrates part of chromosome segments from the *Brassica rapa* genome through interspecific hybridization with *B. rapa* [15]. Moreover, evolutionary studies demonstrated that the genetic diversity of natural populations in two ancestors of rapeseed varies greatly, with higher genetic diversity in the natural populations of *Brassica rapa* than in natural populations of *Brassica oleracea* [36]. With the above facts in mind, it was at first glance strange that there were more *h*-QTLs distributed on the C subgenome than the A subgenome. One possible reason could be that the genetically diverse regions in terms of SNP abundance might not be those functional regions, as there was a biased expression of functional genes between the two subgenomes. Further, the allelic variations on the genetically conserved C subgenome, not those 'wild' alleles on the genetically diverse A subgenome, were more valuable to cause F1 heterosis. The knowledge about the asymmetric distribution of *h*-QTLs suggests the selection of parents with a more allelic variation on C genomes which are valuable for F1 heterosis.

The results do not imply that all forms of heterosis resulted from the *h*-QTLs with low PGSI. On contrary, PGSI should be high, at Chr.C03-No.08, Chr.C04-No.04, Chr.C07-No.02, Chr.C04-05, and Chr.C08-05 to achieve high GY-HPH, NSS-HPH, NSP-HPH, NBP-HPH, and PH-HPH, respectively (S9–S14 Tables). Heterosis has been proposed as an alternative term for 'heterozygosis' to avoid limiting the term to the effects that would only be explained based on heterozygosity according to Mendelian inheritance principles [37]. Heterozygosity between parents does not always give rise to hybrid vigor. Genetic incompatibility between parents could reduce fitness via a form of 'outbreeding depression' [38,39].

We adopted regression coefficients of the variables of the regression models as the criteria for selecting *h*-QTLs. Numerous *h*-QTLs responsible for GY-, TSW-, NSS-, NSP-, NBP-, and PH-HPH, respectively, were identified and illustrated (Figs 4 and S2–S6). The candidate genes responsible for high-impact *h*-QTLs were suggested (S15–S20 Tables). The field experiment with 400 hybrids in three replicates was not a very small scale for rapeseed. Moreover, the CMS pool consisting of maternal lines was genetically diverse, the population that contained 400 hybrids demonstrated a wide range of heterosis in all the six agronomic traits investigated.

In conclusion, we demonstrated in this study the high heterosis of F1 hybrids in semi-winter rapeseed ecotype using the *Ogu*-CMS system and implemented GS-based models for prediction of heterosis and identification of heterotic parental combinations. PGSI negatively influenced GY, NSS, NSP, and PH, but positively influenced TSW. The C subgenome had a greater impact on heterosis than the A subgenome in the polyploidy genome of *B. napus*. We went a step further and drew maps showing overviews of *h*-QTLs across the genome, at which PGSI positively or negatively associated with GY- HPH, TSW- HPH, NSS-HPH, NSP- HPH, NBP- HPH, and PH-HPH, and listed the IDs of the genes underlying *h*-QTLs. Using the GS-based prediction models, combining abilities between an *Ogu*-CMS-pool member and a potential restorer can be tested *in silico* by pairing the nucleotide sequences of parents. Such models could sidestep laborious work, such as testing crosses in fields while facilitating breeding efforts via the transfer of restorer genes to a restorer candidate. The achievements here provide a case of heterosis prediction in polyploid genomes with relatively large genome sizes.

## Materials and methods

### Definition of high and mid parent heterosis

High and mid-parent heterosis were calculated according to the formula below.

$HPH = \frac{F1-HP}{HP}$, where, HPH stands for high parent heterosis; F1 is the phenotypic value of the F1 hybrid; HP represents the phenotypic value of the high parent.

$MPH = \frac{F1-MP}{MP}$, where, MPH stands for mid-parent heterosis; F1 is the phenotypic value of the F1 hybrid; MP means the average phenotypic value of the parents.

### Construction of the *Ogu*-CMS pool

Semi-winter genotypes that represent the genetic diversity of cultivars in the Yangtze River Basin were carefully selected to develop *Ogu*-CMS lines. Genomes of the CMS lines were deeply (30×) sequenced and analyzed. Their genetic diversity was analyzed in the background of a worldwide germplasm collection consisting of 1,057 accessions [39]. After principal component analysis (PCA), 50 CMS lines were selected for the construction of the *Ogu*-CMS pool. PCA was performed using the smartPCA program in the EIGENSOFT package (https://github.com/DReichLab/EIG; v.6.0.1). Different ecotype samples were separated by two principal components (PCs), that is, the winter type was separated from the semi-winter and spring ecotypes by PC1, while the semi-winter type was separated from winter and spring ecotypes by PC2.

### Genome resequencing

DNAs of 50 *Ogura* CMS lines and 10 restorers were extracted and sequenced using a previously described method [15]. Genomic DNA was extracted from young leaves using a cetyltrimethylammonium bromide-based protocol. A NanoDrop2000 spectrophotometer (Thermo Fisher Scientific) was used to determine the quality and concentrations of the genomic DNA. DNA libraries were constructed for each line for Illumina sequencing (Illumina, California, USA) according to the manufacturer's (Biomarker Technologies Cooperation, Beijing, China) instructions. Following DNA-library construction, the accessions were resequenced on an Illumina HiSeq XTen (Illumina, California, USA) platform using a commercial service, with a 150-bp read length. In total, 2,862-Gb high-quality sequences were obtained. All clean reads were mapped to the 'Darmor-*bzh*' reference genome [13], resulting in a 38-fold coverage and a 99.2% mapping rate on average. SNPs and InDels within the 60 accessions were called using

the HaplotypeCaller module in GATK [40] and were filtered based on parameters applied in a previous study [15].

## Definition of parental genetic similarity index

The PGSIs of each cross were calculated using 1-Mb, 500-Kb, and 100-Kb window widths, and the entire genome could be divided into 873, 1722, and 8522 blocks, respectively, based on the window widths. For each block, the sites where parents had the same single nucleotides were marked as '2'. The sites where one parent had a nucleotide similar to the reference but the other parent had a different nucleotide were marked as '1'. The sites where both parents had different nucleotides from the reference were marked as '0'. The PGSI of a block was two times the value obtained by accumulating the marks and then dividing them by the number of loci available for computation.

## Phenotyping and phenotypic data analysis

The 50 *Ogu*-CMS lines were used as female parents, and the eight restorer lines were used as male parents to produce 400 hybrids based on an incomplete double-cross design. Another 100 hybrid lines were produced between the *Ogu*-CMS-pool members and two restorers, independent of the 400-hybrid training population. The training and testing populations were grown in Zhangye, Gansu Province (100˚85E, 38˚43N) in 2017, Hangzhou, Zhejiang Province (120˚19E, 30˚26N) in 2018, and Huzhou, Zhejiang Province (119˚91E, 30˚01N), in 2019. The phenotype values of six agronomic traits including PH, NBP, NSP, NSS, TWS, and GY were measured. The experiments were based on a randomized-complete-block design with three replicates. At least three plants were sampled for each genotype in each replicate. For the NSS trait, 30 siliques from the main inflorescences of each plant were harvested and counted to determine the number of seeds in each silique.

To facilitate the subsequent analyses, phenotypic values from the three environments were integrated according to a linear mixed model as follows: $y = 1_r\mu + Zg + Eu + e$, where $y$ is the vector of the mean value of each genotype in each environment calculated in the first step; $r$ is the sum of the number of genotypes measured in each environment and $1_r$ is an $r$-dimensional vector of 1's; $\mu$ is the common intercept; $g$ is the vector of genotypic effects of all genotypes and $Z$ is the corresponding design matrices for $g$; $u$ is the vector of environmental effects and $E$ is the corresponding design matrix for $u$. The genotypic effect $g$ was assumed to be a fixed value to gain the best linear unbiased estimation (BLUE) of each genotype across environments. The BLUE value of each genotype was used to perform all the analyses in the study. All linear mixed models were implemented using the lme4/R program [41].

## Prediction methods

Two parametric methods, GBLUP and LASSO, were applied to predict heterosis. The general model of the two parametric methods that include all $m$ markers is described as follows: $y = X\beta + \sum_{k=1}^{m} Z_k \gamma_k + \varepsilon$, where $y$ is an $n \times 1$ vector of the phenotypic values for each trait; $n$ is the individual size; $X$ is an $n \times q$ matrix of predictors used to predict y; $q$ is the number of predictors in the model; $\beta$ is a $q \times 1$ vector of model effects, $\varepsilon$ is an $n \times 1$ vector of residual errors with an assumed $N(0, I\sigma^2)$ distribution; $Z_k$ is a column for the genotype indicator variable of all $n$ individuals for marker $k$; $\gamma_k$ is the additive genetic effect of marker $k$. The marker $k$ for individual $j$ (where $j = 1, 2, \ldots, n$) in the study is defined as 1, 0, and -1 for homozygote of the minor allele, heterozygote, and homozygote of the major allele, respectively.

The GBLUP method assumes $\gamma_k \sim N\left(0, \frac{1}{m}\phi^2\right)$, where $\phi^2$ represents the polygenic variance shared by all markers. The expectation of $y$ is $E(y) = X\beta$. The variance-covariance matrix is var $(y) = V = K\phi^2 + I\sigma^2 = (K\lambda + I)\sigma^2$, where $\lambda = \phi^2/\sigma^2$ is the variance ratio and $K = \frac{1}{m}\sum_{k=1}^{m} Z_k Z_k^T$ is a marker-generated kinship matrix. The GBLUP method exploits the genomic relationships between training populations and testing populations to predict the genomic values for unknown individuals without estimating marker effects. The GBLUP was implemented using the predhy/R program [11].

The LASSO method assumes $\beta_k \sim N(0, \phi_k^2)$ and $\phi_k^2 \sim \text{Exp}\left(\frac{1}{2}\lambda^2\right)$ for all $k = 1, \ldots, q$, where $\lambda$ is a shrinkage parameter. The method directly estimates marker effects in the training population and predicts the genomic values of individuals in the testing population. When performing LASSO, the marker $k$ for individual $j$ (where $j = 1, 2, \ldots, n$) was defined as PGSI instead of a single nucleotide marker value. Since the number of SNP markers had little effect on the accuracy of the genomic prediction [11], 0.05% of all SNP markers were randomly selected when SNP markers were used as genetic information in LASSO and GBLUP models. The LASSO was implemented using the glmnet/R program in the present study [42]. Since the LASSO method can achieve variable selection, the variables obtained using the LASSO method were extracted and re-estimated using linear regression, and they were implemented in the R program using the default functions. The Pearson correlation coefficient between the observed and predicted heterosis was used to calculate predictability. We provided the codes in getting the predictability as S1 Data.

## Models applied for the prediction of heterosis

GBLUP_A, GBLUP_AD, LASSO_SNP, LASSO_100Kb, LASSO_500Kb, and LASSO_1Mb models were applied to predict the heterosis of six agronomic traits. The two GBLUP models differ from each other in building the models merely based on the consideration of additive effects (GBLUP_A) only or both additive and dominant effects (GBLUP_AD). Conversely, the four LASSO models applied differ based on the units used to calculate PGSIs, which ranged from single nucleotide (LASSO_SNP) to decreased nucleotide sizes, including 100Kb (LASSO_100Kb), 500Kb (LASSO_500Kb), and 1M (LASSO_1Mb) nucleotide fragments. Since the application of the different prediction models resulted in different heterosis predictabilities for the same trait, we adopted the model with the highest predictability for a particular trait to predict the heterosis of a trait. For example, the LASSO_100Kb and LASSO_500Kb models were selected for the prediction of GY-HPH and PH-HPH, respectively.

## Predictability drawn from ten-fold cross-validation

Predictability was drawn from 10-fold cross-validation, in which nine parts of a sample were used to estimate parameters used for the prediction of heterosis in the remaining part of the sample. Eventually, each individual was predicted once and used nine times to estimate the parameters. The Pearson correlation coefficient between the observed and predicted heterosis was used to calculate predictability. We replicated the cross-validation analysis 100 times, and the predictability of each trait was the average value of the 100 times prediction.

## Verification of prediction model

A testing population containing 100 hybrid lines was developed by crossing the *Ogu*-CMS-pool members with two independent restorers that were not used to calculate the predictabilities of the training population. The optimal model for NSS-HPH, NSP-HPH, NBP-HPH, and PH-HPH was LASSO_100Kb, and the optimal models for TSW-HPH and GY-HPH were

LASSO_500Kb and LASSO_1MB, respectively (Table 1). LASSO_100Kb was adopted to predict NSS-HPH, NSP-HPH, NBP-HPH, and PH-HPH. LASSO_500Kb and LASSO_1MB were adopted to predict TSW-HPH and GY-HPH, respectively. The optimal model for GY-MPH, TSW-MPH, NSP-MPH, and PH-MPH was LASSO_100Kb. The optimal models for NSS-MPH and NBP-MPH were LASSO_500Kb and LASSO_1MB, respectively (Table 2). Therefore, LASSO_100Kb was adopted to predict GY-MPH, TSW-MPH, NSP-MPH, and PH-MPH, LASSO_500Kb, and LASSO_1Mb were adopted to predict NSS-MPH and NBP-MPH. The predicted and actual values observed in the fields were analyzed to determine the predictability, which indicated the validity of the prediction models.

### Definition of *h*-QTLs and high-impact *h*-QTLs

The regression coefficients of the variables in the model were the criteria for selecting *h*-QTLs. All the variables were defined as *h*-QTLs, and the top 10% and bottom 10% regression coefficients of the variables were considered high-impact *h*-QTLs. Excluding the *h*-QTLs on randomly piled scaffolds, whose positions on certain chromosomes are unknown, all the other *h*-QTLs for specific traits were displayed on the maps, and the underlying genes responsible for high-impact *h*-QTLs were investigated.

### The naming of *h*-QTLs

The name of an *h*-QTL indicates its position on a chromosome. The position of the chromosome from the top to the bottom corresponds to its position from the beginning to the end. The *h*-QTLs of each chromosome were named based on the IDs of chromosomes and series numbers. For example, Chr.C01-01 indicates the No. 01 *h*-QTL, counting from the top to the bottom on Chromosome C01. We indicated positive and negative *h*-QTL on the maps for specific traits. A positive effect shown with a circle on maps means the smaller the PGSI, the great the heterosis, whereas, a negative effect tagged with a triangle means the bigger the PGSI, the greater the heterosis. The gradation of color (dark or tint) represents the degree of an effect.

### Drawing of *h*-QTL map

The *h*-QTL map was drawn by using the RIdeogram/R program [43]. The density of genes on chromosomes is plotted from the annotation file (Brassica_napus.annotation_v5.gff3.gz, https://www.genoscope.cns.fr/brassicanapus/data/).

### Linkage disequilibrium analysis

We used a previously described method for linkage disequilibrium analysis [15]. Briefly, PLINK software (www.cog-genomics.org/plink2; v1.9) was used to calculated complete and partial LD between each pair of SNPs. The squared correlation coefficient ($r^2$) values and the significance of all detected LD between polymorphic sites (P< 0.05) were analyzed for all chromosomes with a 1000-kb window.

### Supporting information

**S1 Fig. Principal component analysis (PCA) plot of the first two components (PC1 and PC2) of the 1057 accessions.** PC1 accounts for 11.19% of the total variation in the winter-type accessions compared to the other accessions, whereas PC2 accounts for 6.90% of the total variation between the semi-winter type and the spring type accession. Green dots represent spring ecotype, blue dots represent winter ecotype, grey dots represent semi-spring ecotype, and red

dots represent sterile lines that were used as female parents in the study.
(TIF)

**S2 Fig. *h*-QTLs responsible for TSW-HPH.** The circles represent the *h*-QTLs that positively contributed to the TSW-HPH, and the triangles represent *h*-QTLs that were negatively correlated to TSW-HPH. The darker the colors of circles and triangles, the greater the effects of the *h*-QTLs, either positive or negative. The colors on the chromosomes indicate the density of genes. The darker the blue, the lower the gene density, the darker the red, the higher the gene density. A and C stand for the two sub-genomes of *Brassica napus*. A limited number of *h*-QTLs on randomly piled contigs, whose positions on certain chromosomes were unknown, are not shown on the map. A positive effect indicated with a circle on maps means the smaller the PGSI, the great the heterosis, whereas, a negative effect tagged with a triangle means the bigger the PGSI, the greater the heterosis.
(TIF)

**S3 Fig. *h*-QTLs responsible for NSS-HPH.** The circles represent the *h*-QTLs that positively contributed to the NSS-HPH, and the triangles represent *h*-QTLs that were negatively correlated to NSS-HPH. The darker the colors of circles and triangles, the greater the effects of the *h*-QTLs, either positive or negative. The colors on the chromosomes indicate the density of genes. The darker the blue, the lower the gene density, the darker the red, the higher the gene density. A and C stand for the two sub-genomes of *Brassica napus*. A limited number of *h*-QTLs on randomly piled contigs, whose positions on certain chromosomes were unknown, are not shown on the map. A positive effect indicated with a circle on maps means the smaller the PGSI, the great the heterosis, whereas, a negative effect tagged with a triangle means the bigger the PGSI, the greater the heterosis.
(TIF)

**S4 Fig. *h*-QTLs responsible for NSP-HPH.** The circles represent the *h*-QTLs that positively contributed to the NSP-HPH, and the triangles represent *h*-QTLs that were negatively correlated to NSP-HPH. The darker the colors of circles and triangles, the greater the effects of the *h*-QTLs, either positive or negative. The colors on the chromosomes indicate the density of genes. The darker the blue, the lower the gene density, the darker the red, the higher the gene density. A and C stand for the two sub-genomes of *Brassica napus*. A limited number of *h*-QTLs on randomly piled contigs, whose positions on certain chromosomes were unknown, are not shown on the map. A positive effect indicated with a circle on maps means the smaller the PGSI, the great the heterosis, whereas, a negative effect tagged with a triangle means the bigger the PGSI, the greater the heterosis.
(TIF)

**S5 Fig. *h*-QTLs responsible for NBP-HPH.** The circles represent the *h*-QTLs that positively contributed to the NBP-HPH, and the triangles represent *h*-QTLs that were negatively correlated to NBP-HPH. The darker the colors of circles and triangles, the greater the effects of the *h*-QTLs, either positive or negative. The colors on the chromosomes indicate the density of genes. The darker the blue, the lower the gene density, the darker the red, the higher the gene density. A and C stand for the two sub-genomes of *Brassica napus*. A limited number of *h*-QTLs on randomly piled contigs, whose positions on certain chromosomes were unknown, are not shown on the map. A positive effect indicated with a circle on maps means the smaller the PGSI, the great the heterosis, whereas, a negative effect tagged with a triangle means the bigger the PGSI, the greater the heterosis.
(TIF)

**S6 Fig. *h*-QTLs responsible for PH-HPH.** The circles represent the *h*-QTLs that positively contributed to the PH-HPH, and the triangles represent *h*-QTLs that were negatively correlated to PH-HPH. The darker the colors of circles and triangles, the greater the effects of the *h*-QTLs, either positive or negative. The colors on the chromosomes indicate the density of genes. The darker the blue, the lower the gene density, the darker the red, the higher the gene density. A and C stand for the two sub-genomes of *Brassica napus*. A limited number of *h*-QTLs on randomly piled contigs, whose positions on certain chromosomes were unknown, are not shown on the map. A positive effect indicated with a circle on maps means the smaller the PGSI, the great the heterosis, whereas, a negative effect tagged with a triangle means the bigger the PGSI, the greater the heterosis.
(TIF)

**S7 Fig. Genome-wide average LD decay in the sterile lines, restore lines, and all lines.** The green, red, and blue curves display the rate of LD decay over distance(Kb) in all sixty parental lines, sterile lines, and restore lines, respectively.
(TIF)

**S1 Table. Information of the sixty parental lines involved in the study.**
(XLSX)

**S2 Table. Phenotypic values for all agronomic traits in the three environments.**
(XLSX)

**S3 Table. The genetic relationship between parents and offspring.**
(XLSX)

**S4 Table. The cross IDs of the top 10% for each agronomic trait.**
(XLSX)

**S5 Table. Correlation coefficients between the phenotypic values of the parents and the hybrids.**
(XLSX)

**S6 Table. Correlation coefficients among the six agronomic traits of the hybrids.**
(XLSX)

**S7 Table. Correlations between the PGSI and the heteroses.**
(XLSX)

**S8 Table. Absolute values of the correlation coefficients between the PGSI and the heteroses.**
(XLSX)

**S9 Table. Positions of the *h*-QTLs that are associated with the GY-HPH.**
(XLSX)

**S10 Table. Positions of the *h*-QTLs that are associated with the TSW-HPH.**
(XLSX)

**S11 Table. Positions of the *h*-QTLs that are associated with the NSS-HPH.**
(XLSX)

**S12 Table. Positions of the *h*-QTLs that are associated with the NSP-HPH.**
(XLSX)

**S13 Table. Positions of the *h*-QTLs that are associated with the NBP-HPH.**
(XLSX)

**S14 Table. Positions of the *h*-QTLs that are associated with the PH-HPH.**
(XLSX)

**S15 Table. Genes underlying the *h*-QTLs for GY-HPH.**
(XLSX)

**S16 Table. Genes underlying the *h*-QTLs for TSW-HPH.**
(XLSX)

**S17 Table. Genes underlying the *h*-QTLs for NSS-HPH.**
(XLSX)

**S18 Table. Genes underlying the *h*-QTLs for NSP-HPH.**
(XLSX)

**S19 Table. Genes underlying the *h*-QTLs for NBP-HPH.**
(XLSX)

**S20 Table. Genes underlying the *h*-QTLs for PH-HPH.**
(XLSX)

**S21 Table. Analyses of variances of predictability from 6 × 2 × 2 × 5 factorial design with six traits, two heteroses (HPH and MPH), two models (GBLUP and LASSO), and five methods (GBLUP_A, GBLUP_AD, LASSO_1Mb, LASSO_500Kb, and LASSO_100Kb).**
(XLSX)

**S22 Table. Comparison of the predictability for heterosis among testing population performed by LASSO using different windows drawn from tenfold cross-validation.**
(XLSX)

**S1 Data. Code to build model and obtain the predictability of the model.**
(DOCX)

## Acknowledgments

We thank Jiangsu Collaborative Innovation Center for Modern Crop Production, China for many helps.

## Author Contributions

**Conceptualization:** Shuijin Hua, Lixi Jiang.

**Data curation:** Yang Xu.

**Formal analysis:** Qian Wang, Tao Yan, Yang Zhu, Dezhi Wu, Yang Xu.

**Funding acquisition:** Xiaoyang Chen, Lixi Jiang.

**Investigation:** Qian Wang, Tao Yan, Zhengbiao Long, Ying Xu, Xiaoyang Chen, Haksong Pak, Jiqiang Li, Shuijin Hua.

**Methodology:** Qian Wang, Luna Yue Huang, Yang Xu, Shuijin Hua.

**Project administration:** Lixi Jiang.

**Resources:** Shuijin Hua.

**Software:** Luna Yue Huang, Yang Xu.

**Supervision:** Lixi Jiang.

**Validation:** Qian Wang.

**Writing – original draft:** Qian Wang.

**Writing – review & editing:** Lixi Jiang.

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
