## [Decision Letter · Decision Letter 0]

25 Jul 2021

Dear Dr Jiang,

Thank you very much for submitting your Research Article entitled 'Prediction of Heterosis in the Recent Rapeseed ( Brassica napus ) Polyploid by Pairing Parental Nucleotide Sequences' to PLOS Genetics.

The manuscript was fully evaluated at the editorial level and by three independent peer reviewers. The reviewers appreciated the attention to an important problem, but raised some substantial concerns about the current manuscript. Based on the reviews, we will not be able to accept this version of the manuscript, but we would be willing to review a much-revised version. We cannot, of course, promise publication at that time.

If you decide to revise the manuscript for further consideration at PLOS Genetics, please aim to resubmit within the next 60 days, unless it will take extra time to address the concerns of the reviewers, in which case we would appreciate an expected resubmission date by email to plosgenetics@plos.org.

[LINK]

We are sorry that we cannot be more positive about your manuscript at this stage. Please do not hesitate to contact us if you have any concerns or questions.

Yours sincerely,

Zhixi Tian, Ph.D

Associate Editor

PLOS Genetics

Li-Jia Qu

Section Editor: Plant Genetics

PLOS Genetics

Reviewer's Responses to Questions

**Comments to the Authors:**

Reviewer #1: Heterosis is very important in agricultural productions. This work addressed yield heterosis in rapeseed by crossing 50 CMS accessions and 8 restorers. The population was then used for genomic selection. The work may provide useful resources for future heterosis sutdies. Below are suggestions for the work:

1. Actually, the sample size (n=400) is not enough for yield traits. Any discussions?

2. GO analyses should be removed. Such analyses applied in this paper are misleading! In one interval, there are tens of genes, and only one of them are the causal one. Pooling tens of "false" genes and a correct one for GO must lead to wrong conclusions.

3. No independent validation crosses for the modelling. The authors should generate tens of crosses (not from 50 CMS accessions and 8 restorers) and tested the accuracy of their GS model.

4. The genotypic and phenotypic data should be publically available for authors.

Reviewer #2: Wang and his/her many colleagues made efforts to predict heteroisis of rapeseed by pairing parent nucleotide sequences. They developed prediction models by introducing the concept of regional parental genetic-similarity index (PGSI) and successfully reduced dimension in the calculation matrix to give more precise prediction. Moreover, they identified heterosis-QTLs and partitioned the impact of heterosis per subgenome and chromosome. They described a useful approach which is validated through the comparison of field observations and in silico predictions. There are some quite interesting points of this paper which was overall clear and well presented.

The authors concluded that the diversity of C subgenome was more important for the rapeseed heterosis than that of A subgenome. I doubted this conclusion. To the best of my knowledge, the A subgenome was much more genetic diverse than the C subgenome. I would, therefore, expect that the A subgenome would be more important for the heterosis than the C genome. The explanation from the authors would be very much appreciated.

I would also list some specifics for your consideration in the revision.

1. Please mention in the M&M how you calculate high and mid parent heterosis, although I know you defined them in the introduction.

2. Please discuss 10-fold cross-validation. Were there common parents between the training and validation sets?

3. Please list the cross IDs of the top 10% in a table or supplemental table.

4. Please include model used in the legend of Fig 5.

5. Fig 6: which correlations was significant?

6. You did not consider LD. You should better discuss the justification and consequences of not considering.

Reviewer #3: This manuscript describes a interesting study on the heterosis of six agronomic traits in rapeseed. The experiments were carried out using several prediction models and the method of PGSI may reduce the computational load for prediction in the crops with large genomes. In addition, CMS lines and the restorer can be very useful resource for community. Overall, this manuscript provides meaningful results of the heterosis prediction in rapeseed with polyploid genomes.

There are some comments that the authors may need to address.

Lines 75-77, please add the detail for calculation of the HPH and MPH. For different traits, do you always use the higher values as the better parent for HPH? For some traits, the higher value is not always the expected phenotype, for example, higher plant height may result in logging.

Lines 96-105, the authors mentioned the advantage of GS comparing to MAS, I would expected to discuss somehow in your results. The h-QTLs here are the genomic regions, so it’s really interesting to study how is the situation if only the h-QTLs were used for prediction.

Lines 153-158, please provide the variance explained by the PC1 and PC2, and please use a different color to show the distribution of 50 CMS lines in Figure S1.

Lines 171-175, from the Figure 1, PH in most of hybrids are higher than those in the male parents. As expected, the female parents would affect the hybrids more than the male parents, however the correlations between the hybrids and their male parents were higher. Do you have an explanation or more details about this?

Lines 179-180, may something be wrong in the figure 2B and table S3 -- NPB or NBP, not consisytent with the manuscript text, please check.

Lines 198-202, what’s the hypothesis behind this in studying the influence of HPH regardless of the traits and what is the meaning of the results here?

Line 207, how did you get the h-QTLs? it’s not clear and please provide more details in the method.

Lines 208-225, you defined positive and negative effects and both of them will result in higher GY-HPH. What’s the different between positive and negative effects? If the positive effect will result in a higher GY in hybrids?

Lines 227-235, using PGSI to identify the h-QTLs, it decreases the complexity of computational load. As far as I understand, it also decreases the mapping resolution and results in a large genomic region, which as a result increases the number of genes underline the h-QTLs. Then, what’s the advantage to use a high density of marker?

Lines 238-242, it’s not clear and please rewrite these sentences.

Line 246, 1860 genes! it’s really a lot. Do you think all of these genes have an important influence on the heterosis of PH?

Line 277, the cross-validation only with 10 replications, it’s not enough especially for the traits of NSS. What’s the meaning of A,B,C,D… in Figure 5?

Lines 292-294, How did you code the design matrix of additive and dominance effect in your GBLUP_A and GBLUP_AD model? Usually, the dominance effect is related with heterosis, so what’s the explanation of that no difference between the two model?

Lines 348-356, it’s a nice discussion about the influence of LD distance. It may be valuable to estimate the LD distance in this population and then to calculate PGSI by considering the LD distance, such as 1(LD distance), 2(LD distance) and 3(LD distance) …

Line 385, Figure 1 change to Figure S1?

Lines 524-525, it should be in the results part?

Lines 540-543, were all of the hybrids planted in the 3 years?

Lines 552-553, why not to adjust the effect of replicates in this model directly instead of using the mean value?

Lines 626-633, it’s not clear that how you got the predictability.

**Have all data underlying the figures and results presented in the manuscript been provided?**

Reviewer #1: None

Reviewer #2: None

Reviewer #3: Yes

PLOS authors have the option to publish the peer review history of their article (what does this mean?). If published, this will include your full peer review and any attached files.

Reviewer #1: No

Reviewer #2: No

Reviewer #3: No

---

## [Decision Letter · Decision Letter 1]

15 Oct 2021

Dear Dr Jiang,

We are pleased to inform you that your manuscript entitled "Prediction of Heterosis in the Recent Rapeseed ( Brassica napus ) Polyploid by Pairing Parental Nucleotide Sequences" has been editorially accepted for publication in PLOS Genetics. Congratulations!

Yours sincerely,

Zhixi Tian, Ph.D

Associate Editor

PLOS Genetics

Li-Jia Qu

Section Editor: Plant Genetics

PLOS Genetics

Comments from the reviewers (if applicable):

Reviewer's Responses to Questions

**Comments to the Authors:**

Reviewer #1: I'm satisfied with the revison, no further requirements.

Reviewer #2: The manuscript of this version has been very much improved. The authors performed the calculation again with much more replications (100 instead of 10), and addressed all my concerns.

Reviewer #3: The authors have well addressed my questions/comments and the manuscript was improved much better and satistified to me. I have no further comments.

**Have all data underlying the figures and results presented in the manuscript been provided?**

Reviewer #1: None

Reviewer #2: None

Reviewer #3: Yes

PLOS authors have the option to publish the peer review history of their article (what does this mean?). If published, this will include your full peer review and any attached files.

Reviewer #1: No

Reviewer #2: No

Reviewer #3: No

**Data Deposition**

http://datadryad.org/submit?journalID=pgenetics&manu=PGENETICS-D-21-00874R1

**Press Queries**

---

## [Editor Report · Acceptance letter]

29 Oct 2021

PGENETICS-D-21-00874R1 

Prediction of Heterosis in the Recent Rapeseed ( Brassica napus ) Polyploid by Pairing Parental Nucleotide Sequences 

Dear Dr Jiang, 

We are pleased to inform you that your manuscript entitled "Prediction of Heterosis in the Recent Rapeseed ( Brassica napus ) Polyploid by Pairing Parental Nucleotide Sequences" has been formally accepted for publication in PLOS Genetics! Your manuscript is now with our production department and you will be notified of the publication date in due course.

With kind regards,

Zsofia Freund

PLOS Genetics

On behalf of:
